# Activation of Mast Cells by Neuropeptides: The Role of Pro-Inflammatory and Anti-Inflammatory Cytokines

**DOI:** 10.3390/ijms24054811

**Published:** 2023-03-02

**Authors:** Dorina Lauritano, Filiberto Mastrangelo, Cristian D’Ovidio, Gianpaolo Ronconi, Alessandro Caraffa, Carla E. Gallenga, Ilias Frydas, Spyros K. Kritas, Matteo Trimarchi, Francesco Carinci, Pio Conti

**Affiliations:** 1Department of Translational Medicine, University of Ferrara, 44121 Ferrara, Italy; 2Department of Clinical and Experimental Medicine, School of Dentistry, University of Foggia, 71100 Foggia, Italy; 3Section of Legal Medicine, Department of Medicine and Aging Sciences, G. d’Annunzio University of Chieti-Pescara, 66100 Chieti, Italy; 4Clinica dei Pazienti del Territorio, Fondazione Policlinico Gemelli, 00185 Rome, Italy; 5School of Pharmacy, University of Camerino, 62032 Camerino, Italy; 6Section of Ophthalmology, Department of Biomedical Sciences and Specialist Surgery, University of Ferrara, 44121 Ferrara, Italy; 7Department of Parasitology, Aristotle University, 54124 Thessaloniki, Greece; 8Department of Microbiology and Infectious Diseases, School of Veterinary Medicine, Aristotle University of Thessaloniki, 54124 Macedonia, Greece; 9Centre of Neuroscience of Milan, Department of Medicine and Surgery, University of Milan, 20122 Milano, Italy; 10Department of Morphology, Surgery and Experimental Medicine, University of Ferrara, 44121 Ferrara, Italy; 11Immunology Division, Postgraduate Medical School, University of Chieti, 66100 Chieti, Italy

**Keywords:** mast cell, inflammation, neuropeptide, cytokines, immunity, tumor, allergy

## Abstract

Mast cells (MCs) are tissue cells that are derived from bone marrow stem cells that contribute to allergic reactions, inflammatory diseases, innate and adaptive immunity, autoimmunity, and mental disorders. MCs located near the meninges communicate with microglia through the production of mediators such as histamine and tryptase, but also through the secretion of IL-1, IL-6 and TNF, which can create pathological effects in the brain. Preformed chemical mediators of inflammation and tumor necrosis factor (TNF) are rapidly released from the granules of MCs, the only immune cells capable of storing the cytokine TNF, although it can also be produced later through mRNA. The role of MCs in nervous system diseases has been extensively studied and reported in the scientific literature; it is of great clinical interest. However, many of the published articles concern studies on animals (mainly rats or mice) and not on humans. MCs are known to interact with neuropeptides that mediate endothelial cell activation, resulting in central nervous system (CNS) inflammatory disorders. In the brain, MCs interact with neurons causing neuronal excitation with the production of neuropeptides and the release of inflammatory mediators such as cytokines and chemokines. This article explores the current understanding of MC activation by neuropeptide substance P (SP), corticotropin-releasing hormone (CRH), and neurotensin, and the role of pro-inflammatory cytokines, suggesting a therapeutic effect of the anti-inflammatory cytokines IL-37 and IL-38.

## 1. Introduction

Mast cells (MCs) derive from bone marrow progenitors and after maturation, they migrate into the tissues where they carry out various biological responses, including innate and acquired immunity [1]. In addition, MCs are immune cells involved in a number of disorders including inflammatory, autoimmune, and allergic diseases [2]. The maturation of MCs has been reported to occur in the presence of stem cell factor (SCF), IL-3, IL-4, and IL-9, in vitro and in vivo [3]. MCs are ubiquitous in the human body but are predominantly localized in perivascular tissue, and in the central nervous system (CNS) are located in corticotropin-releasing hormone (CRH)-positive nerve endings [4]. The meninges also possess MCs that can be activated by insults such as stress and toxins, via vascular permeability, an effect that does not occur in MC-deficient rodents [5].

The anti-inflammatory cytokines have provided good results both in vitro and in rodents, generating new therapeutic hopes. However, at the moment, there is no data on clinical treatments in humans, therefore the dosage, efficacy, lifetime and side effects of these cytokines are unknown and further studies are needed to clarify this. The therapeutic targets of MCs are diverse, including antigens such as mutated KIT variants, which have shown promise and are still being studied, but satisfactory studies concerning the inhibition of pro-inflammatory cytokines have not yet been reported.

The goal of this article is to discuss the current status of knowledge on MC and neuropeptide interactions and the role of pro-inflammatory and anti-inflammatory cytokines.

## 2. Mast Cells and Inflammation

Human MCs were first described by Paul Ehrlich, and approximately 100 years ago Gilchrist observed that there was an increase in the number of MCs near the blood vessels. In 1910, Henry Dale, and later, in 1924, Thomas Lewis confirmed these results and reported that MCs released histamine causing wheal reactions and inflammation. In the last ten years, the exponential increase in the number of articles published on MCs testifies to the interest of researchers on this fascinating immune cell. MCs are myeloid lineage cells originating from the bone marrow cells CD34+/CD117+/CD13+ that mature into tissue under the stimulation of growth factors such as SCF. MCs are classic cells of allergic diseases, but they can also mediate angiogenesis, acute and chronic inflammation, autoimmune disorders, tissue repair, neurological diseases, and tumors.

The generation, survival and development of MCs depends on the c-kit receptor which binds SCF by exerting the biological response [6]. c-kit/CD117 is a proto-oncogene transmembrane tyrosine kinase receptor that has been immunolocalized in various cells including MCs. CD117 is a 145-Kd glycoprotein that is the product of the kit-gene, and SCF is the c-kit ligand, named MC growth factor, which promotes autophosphorylation of the c-kit receptor, which mediates signal transduction, critical for the survival of MCs [7]. Moreover, CD117 can be instrumental in the diagnosis of some tumors, such as gastrointestinal ones where MCs are abundant [8]. Experiments in mice have shown that c-kit is encoded at the mouse locus where it affects immature germ cells [9].

However, for ethical and practical reasons, there is not much research on human MCs in vivo and the most significant results on this issue have been obtained from research on rodents. In fact, the biological effects of MCs are studied in MC deficient mice, such as KitW-f/KitW-f, KitW/KitW-v or KitW-sh/KitW-sh, which possess a c-kit receptor dysfunction [10]. 

MCs express various surface receptors that allow them to respond to stimuli such as IL-3, SCF, neuropeptides, and others [11]. Activation of MCs with various compounds triggers the increase in intracellular calcium concentration and plays a key role in various inflammatory diseases, including neurological disorders. The activation of MCs can follow the synthesis of cytokines and chemokines which occurs a few hours after the antigenic stimulus. There are various markers of MCs; among these, we have tryptases, which increase by approximately 10 times after activation of CD63. 

Bacterial products such as lipopolysaccharide (LPS) (a potent macrophage stimulator) also activate MCs, demonstrating that these cells have anti-bacterial functions [12]. The classical high specific activation of MCs occurs through the antigenic reaction with the IgE antibody bound on the cellular high-affinity IgE Fc receptor (FcεRI) [13]. This interaction occurs at high affinity (1 × 10^10^ M^−1^), provoking the aggregation of receptors and eliciting an intracellular biological response [14]. FcεRI is composed of four subunits called alpha, beta and two gammas, responsible for initiating the biological cascade that leads to the generation of proteins that mediate the inflammatory and allergic responses [15]. The beta subunit leads to the amplification of the antigen reaction with the IgE antibody, while the alpha subunit binds IgE and activates the MC [16]. The biochemical cascade of reactions, leading to the transcription and activation of inflammatory cytokines, thus begins with the activation of the FcεRI and the phosphorylation of tyrosine kinases (Src, Syk and Tec family) [17]. Next, the 76 kDa SH2-containing leukocyte protein (SLP-76), the non-T cell activation linker (NTAL), and the phospholipase Cγ (PLCγ) (which regulates calcium in the cell) contribute to recruit protein kinase C (PKC) [18]. The phospholipid PI-4,5-P2 membrane is subsequently hydrolyzed, as well as soluble inositol 1,4,5-triphosphate (IP3), and therefore, diacylglycerol (DAG) is formed [19]. Other signaling reactions include mitogen-activated protein kinase (MAPK), extracellular signal-regulated kinase (ERK), c-Jun N-terminal kinase (JNK), and p38 that lead to the transcription and production of inflammatory cytokines [20,21] (Figure 1a). Literature data on electron microscopy of MCs is increasing significantly as this method, compared with the light microscope, allows to better identify and judge the quality of endo-cellular particles, allowing improved cell morphology and biological study of this interesting immune cell. In Figure 1b,c, MCs are shown magnified with light and electron microscopy, respectively.

MCs are recruited into the inflammatory microenvironment by several chemoattractants, including TNF, which is also produced by MCs. Several chemokines such as CCR2, CCR3, CXCR2, CXCR3, and CXCR4 activate MC receptors that are important for inflammatory cell recruitment. The main cytokine receptors expressed by MCs are IL-1, TNF, IL-3, IL-4, IL-5, IL-6, IL-9, IL-13, INFγ and others, which are also involved in cellular development and inflammation.

Furthermore, the activation of MC-phospholipases leads to the generation of arachidonic acid inflammatory products, such as prostaglandin D2 (PGD2), leukotrienes LTC4, LCD4, and LTE4. PGD2 is an unstable prostaglandin detected as 11β-PGF2α, a more stable compound, that is related to systemic inflammation at non-physiological concentration in peripheral blood [22,23]. Therefore, in the brain, PGD2 mediates inflammation and pain, while LTC4, LCD4, and LTE4 are slow reacting substances of anaphylaxis (SRS-A) involved in asthma and other inflammatory diseases [24,25]. For example, leukotriene E4 is produced by MCs and increases in mastocytosis, fueling inflammation. In fact, the inhibitor of this leukotriene is used in treatment of asthma and shortness of breath [26].

After the MCs have been activated, they degranulate and release the inflammatory mediators stored in their granules in seconds (rapid release) [27]. However, most of the production of the potent inflammatory cytokine tumor necrosis factor (TNF) results from the induction of the corresponding mRNA upon MC activation (late secretion) [28]. Cytokines such as IL-5, IL-6, IL-31 and IL-33, and the chemokines CCL2, CCL5 and CXCL8, are synthesized by MCs through mRNA after activation [29] (Table 1).

Injury in the brain can stimulate pro-IL-1 which is cleaved by caspase-1 to form mature IL-1, which binds its receptor IL-1R in the cell membrane and activates gene expression pathways (Figure 2).

The secretion of IL-1 and TNF in the brain induces fever which is produced in the hypothalamus [30]. The mechanism of the production of fever is not yet fully understood; however, the antigen which can be a neuropeptide and other pyrogens (IL-1, TNF, IL-6, etc.) activates MCs with the production of IL-1 [31,32]. This cytokine acts on the anterior hypothalamus to generate prostaglandin E2 (PGE2), which stimulates the vasomotor center with the heat and fever production [33]. MCs secrete neuropeptides CRH and Substance P (SP), which activate microglia in the brain to generate IL-1 and the chemokines CXCL8 [34]. In fact, MCs cross-talk with microglia by releasing histamine and tryptases which induce the secretion of pro-inflammatory cytokines in MCs [35]. 

IL-6 (also called “myokine”) is involved in fever and is mostly secreted by T lymphocytes, macrophages, and MCs. It stimulates the immune response by binding to its receptor gp130, increasing intracellular calcium and causing a transduction cascade that leads to the activation of the signal transcription factor signal transducer and activator of transcription 3 (STAT3) and MAPK [36]. IL-6 can be secreted by macrophages and MCs in response to specific microbial molecules called pathogen-associated molecular patterns (PAMPs), which bind a group of important receptors of the innate immune system, the pattern recognition receptors (PRRs), to which the Toll-like receptors (TLRs) belong [37]. Furthermore, IL-6 initiates PGE2 synthesis in the hypothalamus, thereby causing an increase in body temperature and mediating systemic inflammation [38]. The relationship between MC-mediated allergic diseases and some neurological pathologies is demonstrated by the higher frequency of allergies and elevated IgE in children with CNS disorders [39]. 

## 3. Neuropeptides

Recent studies exploring the effect of neuropeptides during inflammatory diseases reported that they can stimulate immune cells including MCs to produce pro-inflammatory cytokines that can aggravate the inflammatory clinical picture. In this paper, the study of cytokine inhibition with IL-37 or IL-38, suppressors of IL-1, may result in a new strategy against acute and chronic diseases induced by increased levels of neuropeptides, including substance P, CRH, and neurotensin. Therefore, it is necessary to add more information to the current literature on neuropeptide-induced inflammation and describe new therapeutic strategies, since many inflammatory neurological diseases are incurable.

### 3.1. Substance P (SP)

SP, discovered in 1931 by Von Euler, was characterized in 1970 by Leeman and Chang. It is a highly conserved neuropeptide, isolated from the rat brain, that is mainly secreted by neurons and is involved in nociception, hypotension, muscle contraction, and inflammation [40]. The pro-inflammatory effect of SP, acting through its specific neurokinin-1 (NK-1), was confirmed in a large number of studies [41,42,43].

SP is a member of the tachykinin peptide hormone family, located on human chromosome 7 and encoded by the *TAC1* gene [44,45]. SP activates MCs through specific receptors without degranulation and causes the infiltration of granulocytes through the synthesis of some cytokines such as TNF and IL-8 [46]. In addition, the neuropeptide nerve growth factor (NGF) and NT can activate MCs and participate in inflammatory processes [47]. In fact, activation of MCs, in some neuronal diseases, leads to the activation of NK1 receptors following an increase in vascular permeability [48] (Table 2). 

SP stimulation can cause focal inflammation in the hypothalamus and amygdala, with pathological symptoms in several neurological diseases [49]. Neuropeptides such as SP, CRH, and NT may have a synergistic inflammatory action through the activation of cytokines [50]. Therefore, SP is an important neuropeptide and neuromodulatory compound involved in neurogenic inflammation [51]. It is an activator of several immune cells, including macrophages and MCs, causing pro-inflammatory compounds including interleukins, chemokines and growth factors [31,52]. In the brain, SP is involved in the process of pain reception through the stimulation of the trigeminal nerve. In in vitro studies, SP can activate MCs to secrete pro-inflammatory mediators such as cytokines, chemokines, arachidonic acid products and proteases. All these compounds participate in the local and systemic inflammatory response [53,54]. 

In addition, SP induces vascular endothelial growth factor (VEGF) and IL-33 in a dose-dependent manner in MCs and acts synergistically with them in inflammatory responses [41,55]. The cytokine IL-33, also called “alarmin”, is generated and secreted by MCs and increases the ability of the SP to stimulate MCs to release VEGF, TNF, and IL-1 [56]. This effect of SP on VEGF and TNF production demonstrates a crosstalk between neuropeptides and pro-inflammatory cytokines [55,57]. Moreover, human MCs stimulated with SP and anti-IgE produce IL-33 which increases the release of IL-31, a cytokine involved in atopic dermatitis (AD) and transmitting itch diseases involving the CNS [58]. This demonstrates that SP is a potent pro-inflammatory neuropeptide that can activate and contribute to the secretion of certain pro-inflammatory cytokines [59]. In fact, serum elevations of SP and its analogue hemokine-1, along with serum levels of the cytokines TNF and IL-6, have been detected in inflammatory diseases including fibromyalgia [60]. Additionally, in mouse models, it has been observed that SP stimulates angiogenesis through the proliferation of endothelial cells, causing neurogenesis and peripheral inflammation [61]. 

SP performs its biological action by binding to tachykinin NK1 receptors located on the membrane of vascular endothelial cells, inducing inflammation, vascular permeability and edema [62]. SP mediates pain through the trigeminal nerve and activates MCs to produce inflammatory mediators including leukotrienes and prostaglandins [63].

In some neurological and psychiatric disorders such as depression, bipolar mood disorder and anxiety, where MCs are in contact with SP positive nerves, an increase in SP, TNF and VEGF is observed [64]. However, some neuropeptides, such as epinephrine, a neurotransmitter beta-receptor agonist, are capable of inhibiting TNF, histamine, and PGD2 released by activated MCs, an effect that impedes some diseases including asthma [65]. 

### 3.2. Corticotropin-Releasing Hormone (CRH)

As early as 1948, Harris reported that the hypothalamus is an important connecting organ between the nervous and endocrine systems [66]. Information is generated directly from the nuclei of the hypothalamus to the Locus Coureuleus, the main center for the release of norepinephrine, and from here, an efferent pathway starts that proceeds directly to the adrenal medulla causing the release of catecholamines, including adrenaline, norepinephrine, and dopamine [67,68]. Catecholamines can interact with pro-inflammatory cytokines secreted after cognitive (stress) or non-cognitive (microorganisms) stimuli [69]. Circulating cytokines reach the CNS and bind to their receptors, exerting an inflammatory effect [70]. CRH acts through two receptors, corticotropin-releasing hormone receptor (CRHR)-1 [71] and CRHR-2 [72] which is subdivided into CRHR-2α and CRHR-2β [73].

In 1981, Vale et al. isolated and characterized CRH from the sheep hypothalamus. It is a very similar peptide to the human one and is widely expressed in the brain [74]. The hypothalamus secretes CRH, allowing the pituitary gland to release adreno-corticotrophic hormone (ACTH) which, at the level of the cortex of the adrenal glands, will allow the release of cortisol, an anti-inflammatory marker [75,76]. Cortisol and catecholamines produced by the adrenal gland act at an anti-inflammatory level by inhibiting pro-inflammatory cytokines such as IL-1, TNF and IL-6 [77]. The parvocellular neurons of the paraventricular nucleus are the major source of CRH in rats and humans [78]. This hormone is secreted and then transported to the anterior part of the pituitary gland [79]. 

In humans, CRH can also be found outside the CNS, such as in the adrenal medulla, stomach, placenta, pancreas, duodenum, and in some tumors [80]. The administration of CRH in vivo causes an elevation of ACTH and cortisol in plasma, with side effects depending on the administered dose [81]. Thus, CRH is secreted by the hypothalamus after antigen stimulation and activates the hypothalamic–pituitary–adrenal (HPA) axis, but it can also be released from nerves outside the CNS and produced by immune cells including MCs [82]. 

Immune cells such as MCs and other cells are interconnected to the CNS through the production of cytokines influencing the physiological behavior of the body [83]. CRH secretion increases during stress and may lead to hypercortisolism. Moreover, CRH can promote MC maturation and induces neurogenic inflammation [84]. In fact, the histamine produced by MCs increases CRH both at the protein level and by inducing mRNA [29,85]. MCs can produce and be activated by neuropeptides which include SP, CRH, NGF and neurotensin (NT), an effect that can be altered by cytokines [86] (Table 3). 

Some CNS diseases can lead to an increase in the expression of CRH-1 receptor in MCs [87], with an elevation of CRH levels and the generation of neurological disorders. In some neurological disorders, there may be an increase in MCs expressing the CRH-1 receptor with induction of inflammation [88,89].

The secretion of IL-1 and IL-6 by MCs also causes stimulation of the CRH that can further stimulate the release of VEGF [90] and PGE2α [91]. 

### 3.3. Neurotensin (NT)

In 1973, Susan Leeman (today a leading researcher in our group) isolated NT for the first time. NT is a 13 amino acid peptide which plays the role of neuromodulator and neurotransmitter in the CNS [92]. There are four cell receptors that bind NT: NTSR 1, NTSR2 and the type 1 receptors Sortilin 1 (Sort 1) and SorLA [93]. The NTSR1, with high neuron affinity, and the NTSR2, with low activity, were discovered first and therefore they are the most studied [94]. NTSR1 is expressed more in neurons, while NTSR2 is poorly expressed, but these data still need to be confirmed [95]. NT regulates the digestive tract and cardiovascular system and is a mediator of neurological responses such as pain, psychosis, temperature regulation, sensitivity to ethanol, analgesia, etc. [96]. Some studies show that NT can act as an antipsychotic-induced dopamine against schizophrenia without modifying NTS1 receptors [97]. It has also been observed that low levels of NT in the thalamus lead to a stimulation of greater alcohol consumption [98]. 

NT is mostly found in the brain and gut where it becomes involved with inflammatory processes [99]. It induces hormone secretion from the anterior pituitary gland and when it is administered in mice, causes side effects such as antinociception and hypothermia [100]. NT secreted under stress acts synergistically with CRH to stimulate MCs, resulting in increased vascular permeability, generation of VEGF, and disruption of the blood–brain barrier (BBB) [101]. This tridecapeptide which impacts the brain and other organs is fundamental in some inflammatory processes [102]. 

In addition, NT induces the expression of CRHR-1 and CRH protein which stimulates MCs in allergic diseases [103]. In an interesting article, we reported that NT stimulates the gene expression and release of IL-1β and CXCL8 from cultured human microglia, underlining that the NT is a proinflammatory peptide [104] and therefore confirming that NT plays an important role in inflammation. Immune cells such as lymphocyte macrophages and MCs are activated by NT with secretion of inflammatory cytokines and B cell immunoglobulin production, an effect that emphasizes the crosstalk of NT with the immune system [105]. At the cellular level, NT increases calcium levels and the production of nitric acid, a pentavalent nitrogen oxyacid, with strong oxidizing power. NT is therefore a brain neurotransmitter that interacts with MCs, both in innate and acquired immunity, and has a synergistic action with CRH on the stimulation of MCs, activating a neuro-immune mechanism [106]. These effects, which play a stress-mediating role, were observed in vivo in rodents [107,108]. In fact, NT and CRH released in the brain mediate CNS-related inflammatory disorders, such as stress, psoriasis, and AD, where MCs play a crucial role [109]. 

MC activation by neurotransmitters can lead to the release of histamine and increases vascular permeability with the generation of headache [110]. There are several neuropeptides that can activate cerebral MCs that can produce pathological phenomena. For example, SP is capable of inducing itch due to the interaction of SP with neurokinin-1 receptor (NK1R) on neurons. NT binding its receptor NTR1 of the MC, stimulates the production of cytokines and chemokines through the activation of RAS, a signaling protein associated with the cell membrane, with phosphorylation of the *Raf* gene family which activates MEK 1/2 and ERK ½, which promote the transcription factor AP-1 in the nucleus and, consequently, the production of cytokines/chemokines [111,112] (Figure 3). Stress-induced NT can mediate neurogenic inflammation in SP-enhanced mice, causing activation and degranulation of MCs via the NK-1 receptor which induces, in turn, the upregulation of SCF or IL-4, important for proliferation of the MCs [113]. 

In conclusion, NT in the brain causes the activation and proliferation of microglia with brain inflammation, disruption of the blood–brain and the intestinal barrier, along with MC activation and the generation of pro-inflammatory cytokines with neuronal damage [114].

## 4. Anti-Inflammatory IL-37

In the late 1970s, IL-1α and IL-1β were shown to be pro-inflammatory cytokines that we now know can drive inflammation of Th1 and Th17 cells [115]. Inflammatory neuropeptides stimulate IL-1 and other cytokines in the brain. For example, NT stimulates gene expression and secretion of IL-1β and the chemokine CXCL8 in cultured microglia [116]. The IL-1 family includes IL-1α, IL-1β, and other inflammatory cytokines such as IL-18, IL-33, IL-36a, IL-36b, and IL-36γ [117]. Later, other cytokines were discovered, but with anti-inflammatory activity, such as IL-1Ra, IL-37, and later IL-38 and the anti-receptor IL-36Ra. IL-37 is a member of the IL-1 family and is a human anti-inflammatory cytokine, which suppresses innate immunity, modulates acquired immunity, and functions in mice [118]. Five isoform splice variants (“a to e”) of IL-37 have been reported, of which “b” is the most studied. IL-37 is expressed in the nucleus as IL-1 and IL-33, but in contrast to these last two cytokines, IL-37 has anti-inflammatory power, dumping the innate immunity [119]. Activation of the TLR in macrophages leads to the production and secretion of the IL-37 precursor (pro-IL37), which is cleaved by caspase-1 to form mature IL-37. Some IL-37 is transferred into the nucleus, while another part, together with pro-IL-37, is transferred out of the cell, and all are biologically active [120]. In addition, it appears that extracellular proteases may influence pro-IL-37 to become more active. Once generated, IL-37 binds to the IL-18 receptor alpha chain (IL-18Rα), exercising the down-regulation of inflammation [121]. Human monocytes treated with various stimuli such as IL-1 or bacterial LPS are known to produce abundant pro-inflammatory cytokines, an effect enhanced in genetically IL-37 deficient monocytes [122]. These data suggest that IL-37 controls the generation of pro-inflammatory cytokines of the IL-1 family. Indeed, in experimental models such as rheumatoid arthritis, where mice are treated with human IL-37, inflammation is inhibited [123]. The IL-37 gene has only been identified in human cells, but not in mouse cells. However, accurate and sophisticated studies have allowed the generation of transgenic mice expressing human IL-37 (IL-37tg), allowing to better understand the functions and role of this cytokine in the inflammatory process [124]. In these studies, IL-37 confirmed its inflammation-suppressing power even in clinical experimental models [125]. In fact, in mice with inflammatory diseases, such as asthma or ulcerative colitis, treatment with IL-37 improved their pathological condition [126]. IL-37 not only suppresses inflammation and innate immunity, but also plays a regulatory role in acquired immunity by acting on the inhibition of antigen-stimulated T lymphocytes [127]. Thus, IL-37 is produced to protect the body’s tissues against pro-inflammatory stimuli [128]. A number of human cells, including immune cells (such as monocytes, macrophages, B cells, dendritic cells, etc.) express IL-37. Upon stimulation, immune cells treated with IL-37 in vitro produce less IL-1β, IL-6, and TNF [129]. In addition, the migration of inflammatory cells is also inhibited by IL-37, an effect that confirms its anti-inflammatory power. However, the inhibitory mechanisms of IL-37 still need to be clarified, although some authors have hypothesized that IL-37 inhibits mammalian target of rapamycin (mTOR) [130].

IL-37 is generated as a protective, anti-inflammatory effect in the CNS in patients with ischemic stroke and other inflammatory diseases [131]. Therefore, the inhibition of IL-1 family proinflammatory cytokines requires the IL-1 family decoy receptor (IL-1R8) [120]. 

In our recent studies, we reported that IL-37 plays a fundamental role in autism spectrum disorder where it enhances and inhibits NT secretion and gene expression of IL-1 and CXCL8 in microglia [104]. However, the mechanism of action of IL-37 is still unclear even though it is thought that the anti-inflammatory molecule SMAD family member 3 (SMAD3) of the cytokine TGF is involved in the nucleus [127,132]. In light of these studies, we can certainly say that IL-37 opens up therapeutic hope for acute and chronic inflammatory diseases, including autoimmune disorders [133].

## 5. IL-38 Dumping IL-1 Induced Inflammation

IL-1 is the endogenous pyrogen mediator of fever, with activating effects on CNS neurons and therefore is crucial in neurophysiological functions [134]. Increased IL-1 in the brain may result in a number of biological effects, including the elevation of inflammatory mediators and the inhibition of gamma-aminobutyric acid (GABA) receptor responses and calcium fluxes [135]. 

The IL-1 family member genes include pro-inflammatory and anti-inflammatory molecules (Figure 4). The last two cytokines discovered, IL-37 and IL-38, are anti-inflammatory and are located on chromosome 2. IL-38 is thought to be an ancestral gene intended to counteract the pro-inflammatory effects of IL-1 [136]. In humans, IL-38 is detected in several organs such as the tonsils, skin, fetal liver, heart, and placenta, but in future studies, it will certainly be detected in many other organs [137]. The anti-inflammatory effect of IL-38 is homologous with two anti-receptors, IL-36Ra (approximately 40%) and IL-1Ra (approximately 40%), sharing the anti-inflammatory effect [138]. To be active, IL-38 must be processed, but at the moment, the protease(s) responsible for the generation of its mature form are unknown. IL-38 has similar anti-inflammatory potency to IL-37 and IL-36Ra, and works best at low concentrations, while at higher concentrations it may have opposite effects [139]. In vitro studies demonstrated that IL-38 is capable of inhibiting TNF and IL-1 in LPS stimulated THP-1 cells. Moreover, it has been shown that IL-38 used in the truncated form is capable of inhibiting TNF and IL-1 in LPS stimulated THP-1 cells, whereas when IL-38 is used at full length, it can have a stimulatory effect, especially on IL-6. In certain fungal infections, such as Candida albicans, IL-38 suppresses TH17 lymphocytes and inhibits IL-17A, a pro-inflammatory cytokine that is produced in certain immune reactions [140]. The same effect occurred after treatment with IL-36Ra and IL-1Ra [141]. IL-17 comprises six subgroups ranging from IL-17A to IL-17F, and among these, the most studied cytokine is IL-17A [142]. Immune cells and glial cells of the CNS express IL-17A receptors which, once activated, can mediate inflammatory brain disorders [143]. IL-17 generated by γδ T lymphocytes is a cytokine with pro-inflammatory potential that can act synergistically on astrocytes with other cytokines such as TNF, causing the secretion of CXCL1 which recruits neutrophils to the inflammatory site [144]. In addition, in some cases, IL-17 expression may be due to bacterial infections or parasitic infestations which can activate TLR2. In these pathological conditions, treatment with IL-38 could have a therapeutic role [145].

Therefore, it is deduced that the activation of IL-17 in the CNS is very important in inflammatory pathologies [146]. However, in many inflammatory brain disorders where IL-17 is involved, the mechanism of action or whether IL-17 directly causes inflammation is not clear yet, and therefore future studies are needed to clarify this dilemma [147]. 

In light of the results that have been published so far, IL-38 has been shown to play a regulatory role in rheumatic diseases such as rheumatoid arthritis and psoriasis, which are IL-1-mediated diseases [148]. 

Since neuropeptides such as CRH, SP, and others can activate microglia leading to secretion of the proinflammatory cytokines IL-1β, IL-6 and TNF [60], it is pertinent to suppose that inhibiting IL-1 by IL-38 can have a therapeutic effect in inflammatory disorders mediated by IL-1 pro-inflammatory members [138].

## 6. Conclusions

A future use of anti-inflammatory cytokines could replace or be administered in combination with the use of corticosteroids, which in suppressing inflammation cause serious side effects such as immunosuppression. In this article, we investigated the role of the neuropeptide substance P, CRH, and neurotensin in cytokine MC activation. We reported that these neuropeptides stimulate IL-1, IL-6, and TNF. The inhibition of IL-1 by IL-37 or IL-38 could create new therapeutic possibilities and is a new concept that has not yet been reported in the literature for MCs.

MCs, along with macrophages, lymphocytes, endothelial cells, and others, are inflammatory cells that produce IL-1. This cytokine induces other pro-inflammatory cytokines, such as TNF, and possesses an autocrine action by stimulating itself. This effect triggers a cytokine storm with a drastic inflammatory response. Inhibiting IL-1 with IL-37 or IL-38 could be a new therapeutic strategy. However, IL-37 and IL-38 have only been used in research and are not yet available for treatment. Therefore, before using these inhibitory cytokines, certain points need to be clarified such as the concentrations to be used in humans, possible side effects, immunosuppression, and perishability with in vivo treatment.

## Figures and Tables

**Figure 1 ijms-24-04811-f001:**
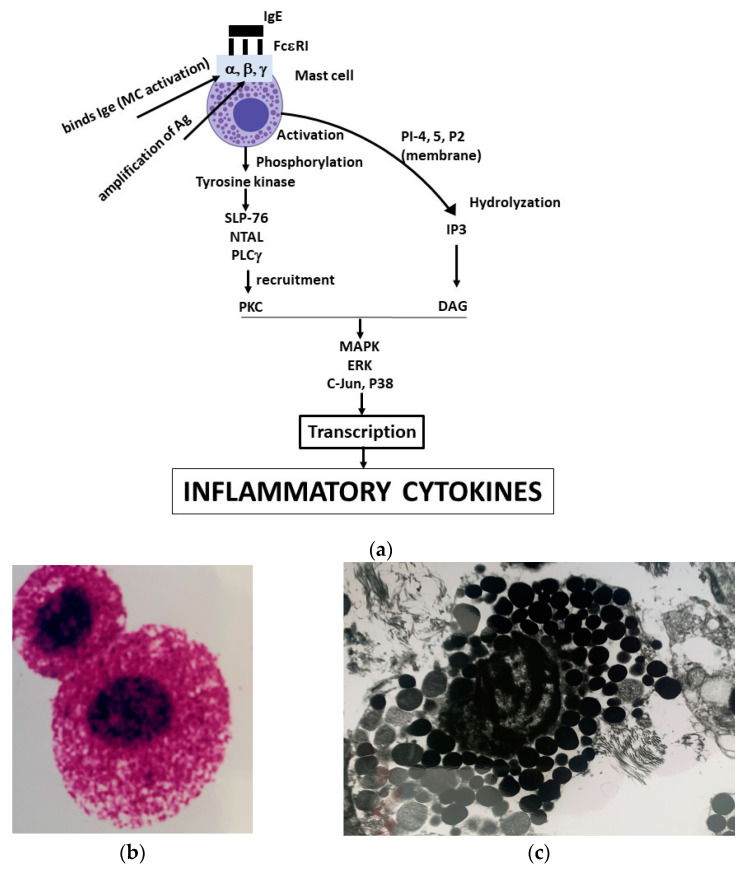
Here, we depict a mast cell (MC) activated by IgE binding the receptor FcεRI, leading a biochemical cascade which results in the transcription and the generation of inflammatory cytokines (panel **a**). Here, we show two MCs magnified 100× with light microscopy (panel **b**), and a MC at electron microscopy, with granules and regular cytoplasm (magnified 30,000×) (panel **c**).

**Figure 2 ijms-24-04811-f002:**
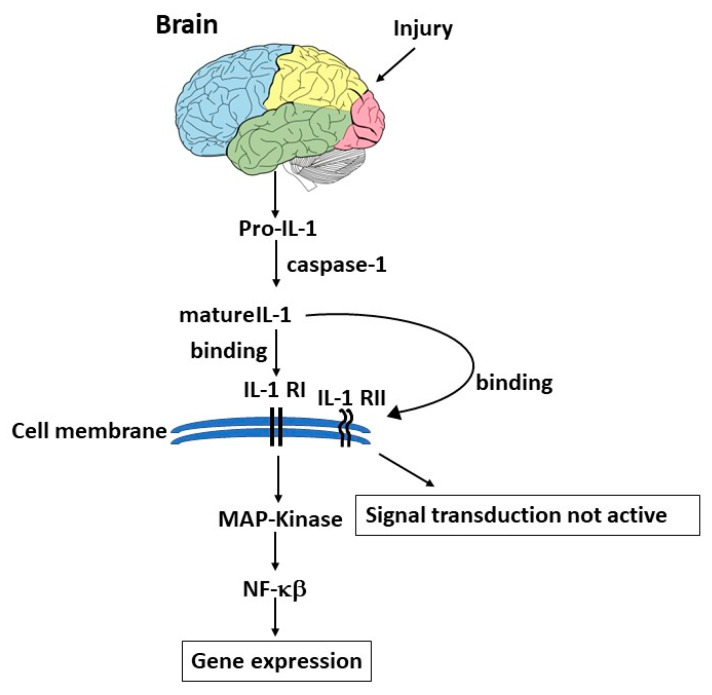
Brain injury causes the activation of pro-IL-1 which is cleaved by caspase-1, leading to mature IL-1 which binds its receptor on the cell membrane, resulting in NF-κB activation and gene expression.

**Figure 3 ijms-24-04811-f003:**
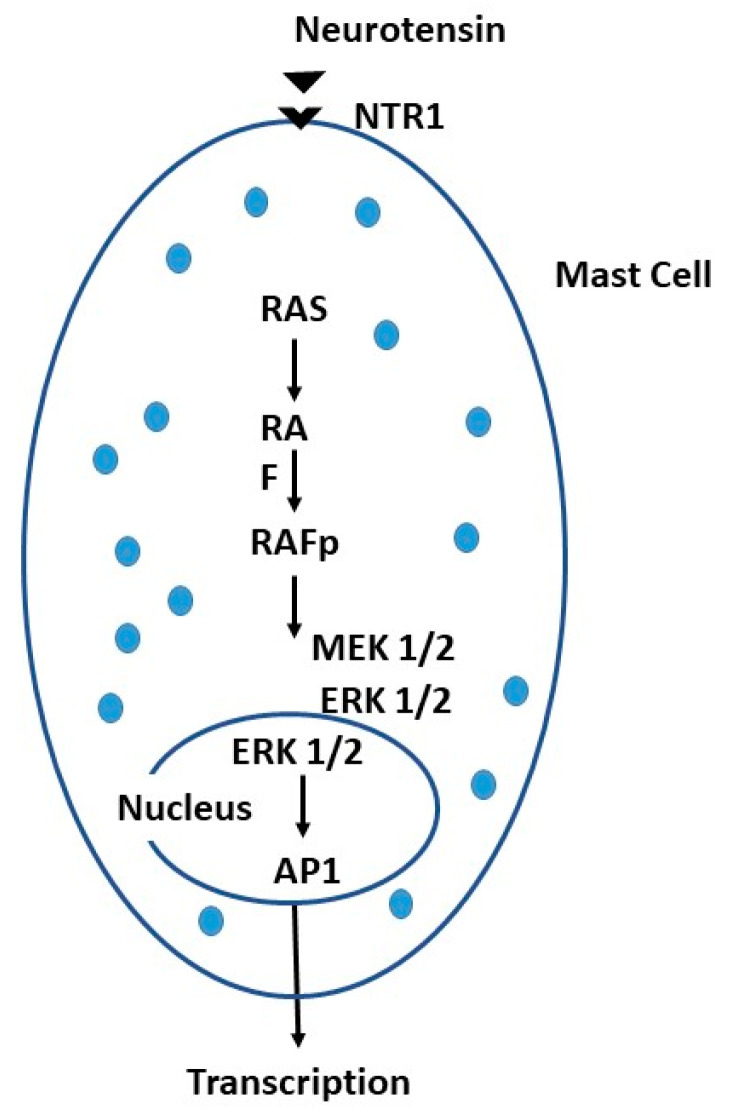
The mast cell (MC) can be activated by neurotensin (NT) through its receptor NTR1, leading to a biochemical cascade in the cytoplasm and nucleus that results in transcription proteins.

**Figure 4 ijms-24-04811-f004:**
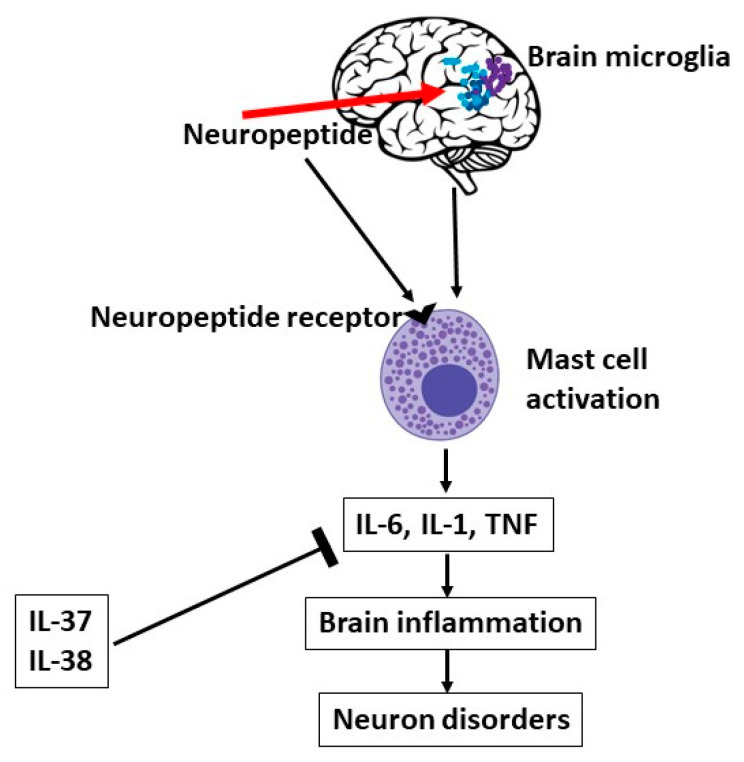
Brain microglia can be activated by neuropeptides, which can also activate mast cells (MCs), leading to pro-inflammatory cytokines, brain disorders, and neuron disorders, an effect that can be inhibited by IL-37 or IL-38.

**Table 1 ijms-24-04811-t001:** Compounds Released by Mast Cells after Activation.

**Compound de novo synthesis:** IL-1, IL-2, IL-3, IL-4, IL-5, IL-6, IL-10, IL-13, TNF, NO, VEGF. **Arachidonic acid products:** prostaglandin PGD2, leukotriene LTB4, LTC4, **Chemokines:** IL-8 (CXCL8), MCP-1 (CCL2), MCP-3 (CCL7), MCP-4, RANTES (CCL5), Eotaxin (CCL11)
**Prestored mediators:** chymase, tryptase, CRH, GM-CSF, SCF, NGF, TGF-β, chondroitin, heparin, histamine, serotonin, β-endorphin, SP, VIP, NT

**Table 2 ijms-24-04811-t002:** Mast Cell Triggers without Degranulation.

**IL-6, TNF or VEGF, NT** (neurotensin), **CRH** (corticotropin releasing hormone), **LPS** (lipopolysaccharide), **VIP** (vasoactive intestinal peptide), **PACAP** (pituitary adenylate cyclase activating polypeptide), **PCBs** (polychlorinated biphenols), **PTH** (parathyroid hormone), **SP** (substance P)
**Heavy metals**: Aluminum, Cadmium, Mercury

**Table 3 ijms-24-04811-t003:** Some peptides released by MCs.

Peptide	Function
Neurotensin	Digestive tract and cardiovascular regulation
Substance P	Inflammation, pain
Kinins (bradykinin)	Inflammation, pain, vasodilation
Corticotropin-releasing hormone	Inflammation: vasodilation
VEGF	Neovascularization, vasodilation
Angiogenin	Neovascularization
Endorphins	Analgesia
Endothelin	Sepsis
Renin	Angiotensin synthesis
Urocortin	Inflammation, vasodilation
Vasoactive intestinal peptide	Vasodilation, mast cell activation

## Data Availability

No new data were created or analyzed in this study. Data sharing is not applicable to this article.

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
