# Peer review of "Activation of Mast Cells by Neuropeptides: The Role of Pro-Inflammatory and Anti-Inflammatory Cytokines"

_ijms, 2023, doi:10.3390/ijms24054811_

Round 1

Reviewer 1 Report

The manuscript of Lauritano and coworkers summarized the role of neuropeptides and cytokines in the mast cells activation. Although of some interest the paper should be thoroughly revised as it appears redundant and poorly focused.  As piece of writing, the literature review must be defined by a guiding concept (e.g. your research objective, the problem or issues you are discussing, or your argumentative thesis). It is not just a set of summaries. A literature review must do these things: be organized around and related directly to the thesis or research question authors are developing; synthesize the results into a summary of what is and is not known; identify areas of controversy in literature and formulate questions that need further research.

The manuscript is also incomplete since the role of several neuropeptides such as VIP, CGRP, Neurotrophins, PACAP, opioid peptides  is lacking or marginally reported.

Conclusions and future directions should also be added.

Reviewer 2 Report

This article explores the current understanding of
MC activation by neuropeptides and the role of pro-inflammatory cytokines, suggesting a therapeutic effect of the anti-inflammatory cytokines IL-37 and IL-38.

The article is very interesting is well written and easy to read. However to complete the review is necessary to introduce the problem of Mast cells in general way considering:

a) Introduction: describe better the problem of mast cells considering the followings points a) Mast cells and evolution b) Histological point, discovery at light and electron microscopy c) Markers of mast cells, c)origin of mast cells d) Functionality of mast cells (receptor, cytokines etc).

b) About the references: there are several very obsolete. Please remove them and insert newer ones

c) May the authors add some pictures of mast cells?

Reviewer 3 Report

In the manuscript entitled “Activation of Mast cells by neuropeptides: The role of pro-inflammatory and anti-inflammatory cytokines” the authors discuss the current status of knowledge on mast cell and neuropeptide interactions. The comments are listed below.

1.    Line# 36: Mast cells are tissue cells…..

A thorough spelling check should be done throughout the manuscript.

2.    The authors should include Conclusion/s section in the manuscript.

3.    What is the link between mast cells and IL-37 and IL-38? What message the authors are trying to convey? The authors should explain in detail.

Author Response

We thank this referee for the good work done on our paper.

-We have corrected the spelling, including the misspelling of mast cell in line #36. (Thank you)

-We have added the conclusion section to the paper, which includes the following as well:

MCs, along with macrophages, lymphocytes, endothelial cells, and others, are inflammatory cells that produce IL-1. This cytokine induces other pro-inflammatory cytokines, such as TNF, and possesses an autocrine action by stimulating itself. This effect triggers a cytokine storm with a drastic inflammatory response. Inhibiting IL-1 with IL-37 or IL -38 could be a new therapeutic strategy. However, IL-37 and IL-38 have only been used in research and are not yet available for treatment. Therefore, before using these inhibitory cytokines, certain points need to be clarified such as the concentrations to be used in humans, possible side effects, immunosuppression, and perishability with in vivo treatment.

Round 2

Reviewer 1 Report

The manuscript has been improved and deserves to be published

Reviewer 2 Report

The authors have answered correctly at all my questions